# Predicting Image Aesthetics for Intelligent Tourism Information Systems

**Ricardo Kleinlein [1],***, **Álvaro García-Faura [1]**, **Cristina Luna Jiménez [1]**, **Juan Manuel Montero [1]**, **Fernando Díaz-de-María [2]** and **Fernando Fernández-Martínez [1]**

[1] Speech Technology Group, Center for Information Processing and Telecommunications, E.T.S.I. de Telecomunicación, Universidad Politécnica de Madrid, Avda. Complutense 30, 28040 Madrid, Spain; agfaura@die.upm.es (Á.G.-F.); cristina.lunaj@upm.es (C.L.J.); juanmanuel.montero@upm.es (J.M.M.); fernando.fernandezm@upm.es (F.F.-M.)

[2] Department of Signal Theory and Communications, University of Carlos III de Madrid, 28911 Leganés, Madrid, Spain; fdiaz@tsc.uc3m.es

* Correspondence: ricardo.kleinlein@upm.es

**Abstract:** Image perception can vary considerably between subjects, yet some sights are regarded as aesthetically pleasant more often than others due to their specific visual content, this being particularly true in tourism-related applications. We introduce the ESITUR project, oriented towards the development of 'smart tourism' solutions aimed at improving the touristic experience. The idea is to convert conventional tourist showcases into fully interactive information points accessible from any smartphone, enriched with automatically-extracted contents from the analysis of public photos uploaded to social networks by other visitors. Our baseline, knowledge-driven system reaches a classification accuracy of 64.84 ± 4.22% telling suitable images from unsuitable ones for a tourism guide application. As an alternative we adopt a data-driven Mixture of Experts (MEX) approach, in which multiple learners specialize in partitions of the problem space. In our case, a location tag is attached to every picture providing a criterion to segment the data by, and the MEX model accordingly defined achieves an accuracy of 85.08 ± 2.23%. We conclude ours is a successful approach in environments in which some kind of data segmentation can be applied, such as touristic photographs.

**Keywords:** computer vision; tourism; domain adaptation; Mixture of Experts; CNN

## 1. Introduction

One of the challenges in computer vision research is the development of models able to make impersonal predictions from subjective data: The response to a given stimulus (either image or video) of an audience when presented with it. For instance, given the visual features of an image, to automatically infer the emotional, cognitive or aesthetic perception of the potential audience. It is known that any visual stimulus has a cognitive effect in the receiver and can induce emotional responses [1,2]. However, the mechanisms that underlie these processes are still largely unknown, making it difficult to find an automatic computational solution to the problem, despite the currently proposed methods [3].

This issue is faced within the frame of the ESITUR project, which narrows down to applications related to tourism. Figure 1 displays the main modules of the general framework. The aim of this project is to develop a system capable of automatically retrieve high-quality audiovisual content for a given location and present it to new users through their smartphones in order to appeal them to visit a given spot. The information displayed (either in a smartphone or in a virtual showcase) is

continuously updated as social media are regularly checked in search of new multimedia content and/or new metadata (GPS coordinates, reviews, comments, number of views, positive or negative reactions, etc.). This collected data is then processed in 2 steps: First, relevant and representative content is identified from both new material uploaded by the users and the reaction to the multimedia content already present in the system. Afterwards, the resources identified as highly representative and relevant to the community are evaluated in terms of their aesthetics value. The aesthetics value here must be understood as a combination of the aesthetics quality and the suitability of a multimedia resource for touristic promotion. A demonstration of a preliminary system working is available in https://youtu.be/lvKL-GD5beM. This way, the introduction of places with touristic interest to potential visitors can be done after an automatic selection of the available content that offers the best possible view of a destination as perceived by most other visiting people. For instance, let us suppose we are weighting two possible touristic routes: Route A includes a place with an aesthetic score of 9 whereas route B has two places with scores 6 and 8 respectively. Option B may be discarded given that the average score is lower than route A's. This ranked content is accessible to the users through mobile devices, but in order to provide relevant material, an accurate selection of those resources is fundamental. Therefore a processing based on how representative a picture is of a given touristic location and on the aesthetics value of the presented audiovisual content is required. In this work we focus on the automatic estimation of the aesthetics of images, adapting experts to homogeneous regions that convey the geolocation of the pictures. We reckon our contribution may help in further smart-tourism image selection work.

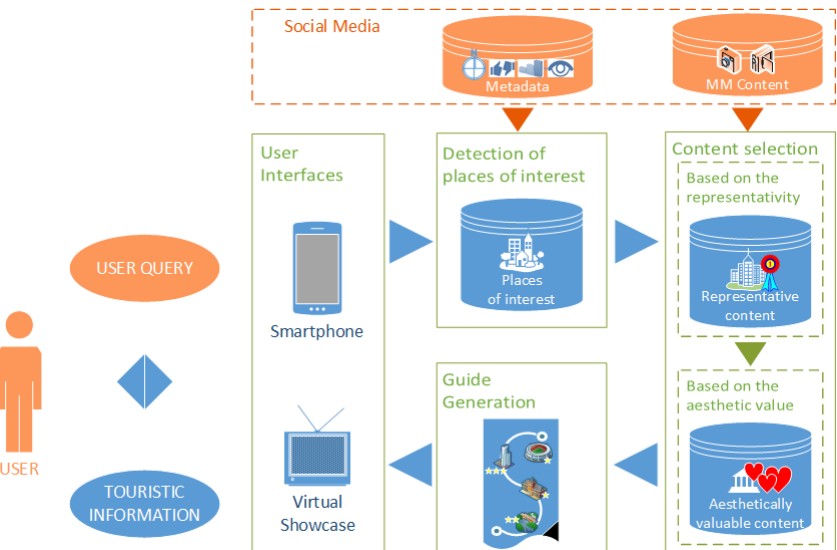

**Figure 1.** Schematic view of the complete framework for the ESITUR project.

The rest of this paper is organized as follows: In Section 2, related work on the topic is summarized. The data used in this study is described in detail in Section 3. The methods for aesthetic value assessment are introduced in Section 4. Section 5 presents the results and experimental setup. We conclude the paper in Section 6.

## 2. Related Work

Aesthetics evaluations are appraisals that arise after stimuli are perceived by the sensory system. Humans are naturally gifted with means to continuously perform these evaluations on sensory information such as audio [4] or images [1,2]. Decisions on the aesthetic value of a stimulus influence many domains of our lives, therefore the desire to replicate these mechanisms with a particular emphasis on visual stimuli since it is the sense humans depend on the most.

The interest in understanding which properties in the visual content drive a human response began studying how different distortions in images such as blurring, noise patching and blockiness could be automatically detected to estimate the perceived quality of an image [5–7]. While both low-level and high-level visual attributes have shown promising results in detecting what makes an image aesthetically pleasant [8–14], lately the use of convolutional neural networks has outperformed any other approach [15–28]. More recently, the intrinsic memorability of images and visual content has also been studied in relation to aesthetics, finding out that unusual or aesthetically pleasant images are negatively correlated to their easiness to be remembered [29,30].

Gainful feature extraction is of uttermost importance in aesthetics assessment, and early work was devoted to a fruitful extraction of low-level features from the visual content. In particular, Datta et al. [8] selected different visual features based on artistic intuition to predict aesthetic and emotional subjective evaluations, whereas Ke et al. [10] considered other features like average hue, contrast, color distribution and the spatial distribution of edges as inputs for a Naïve Bayes classifier. While they opened the path for further research [28], performance from these kind of models quickly showed that an exclusively low-level approach was insufficient for this kind of task [9,14].

Luo and Tang [11] extracted subject regions to compute local features distinguishing between foreground and background. The features they used were clarity, contrast, simplicity, color properties and harmony, lighting and composition geometry. Marchesotti et al [31] and Dhar et al. [12] proposed different sets of generic high-level describable attributes, mainly related to composition and content rules. In Temel and AlRegib [13], they ran an exhaustive evaluation of several high-level descriptors, from geometric (SIFT, GIST, DOG) and color-related (color histogram, hue, color naming) to hybrid approaches, finding that some of these generic features could actually perform just as well as other more complex global descriptors used in previous studies.

Recently, convolutional neural networks have consistently shown better performance than other techniques based on handcrafted features [15–24]. To that aim, different strategies such as using a different loss function [19] or parallel adaptive spatial filters have been studied [16], with promising results that suggest interesting lines of research. Alternative proposals considered the problem of visual aesthetics tightly related to the semantic information contained in the image. For instance, in Mai et al. [15] images are processed by two different networks: The first of them predicts which scene category the input belongs to (human, plant, architecture, landscape, static, animal and night), the latter applies adaptive kernels to local regions of the image. A final aggregation layer combines the output of both networks to predict the aesthethics label of that input image.

In Kao et al. [17], they categorize every image as either an open space scene, an object or a texture pattern picture. For each of these three cases, they train a different specialized convolutional network with different architectures, which process any given input selectively. An analogous procedure is followed in Kao et al. [18], where the prediction of the aesthetics label and the semantics of an image's content is performed at the same time in a multi-task scheme. In all these studies, the semantic classification is driven by knowledge taken from professional photography.

Similar to the idea of dividing the computation among different parts of the system, ensemble models can be thought as a set of submodels in which the global output is given by a weighting of those submodels' [32]. The submodels do not access the same data, but rather specialize in subsets of the training data, enabling the creation of expert models. This mixture of experts can be either cooperative (when the global output comes as a linear combination of the experts' outputs) or competitive, when just one expert evaluates the input and is fully responsible for the global response [33]. Many applications have benefited from this approach since it allows a model to characterize data distributions that can be better understood in terms of more than one regime [34,35].

Our contribution in this paper extends on these systems to the problem of determining the aesthetic relevance of tourism-oriented pictures. Firstly, we adopt a classical approach based on handcrafted visual descriptors fed to a logistic regression model. Secondly, we explore the application of a mixture of experts, all sharing the same architecture, but each specialized in processing images

retrieved from different locations of the target city. To our knowledge, this is the first study exploring the aesthetic value estimation of both closed and open space tourism-oriented images.

## 3. ESITUR Data Collection

The aim of the ESITUR project is to build interactive showcases so pictures and metadata can be retrieved regularly from social media, and attending to the feedback given by the users, update the multimedia content displayed in such showcase. We provide further details of the framework in Appendix 1. In order to carry out an experimentation process as close as possible to the final use case of our model (the development of an interactive showcase that displays multimedia content based on user interaction), we use images from a specific town that could be the subject of an intelligent tourism solution. Therefore, in this section we explain the process followed to obtain aesthetically annotated pictures of that town, shown in general terms in Figure 2.

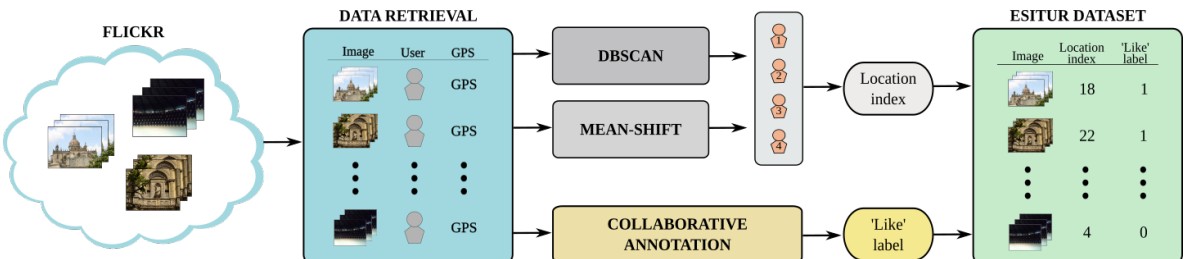

**Figure 2.** Scheme of the data retrieval pipeline. Such data is multimedia content publicly uploaded by the users to a social network(in this case, *Flickr* [36]).

### 3.1. Pictures Retrieval

About 10,000 images featuring different touristic spots in the town of Jerez de la Frontera, Cádiz, Spain, are recovered from *Flickr*, a website that lets its community of users store, find and share amateur pictures or videos online made by themselves [36]. Information regarding the user who uploaded a picture and GPS coordinates of it are also retrieved. Next, two clustering algorithms are executed: one of them is a clustering based on DBSCAN [37] that groups pictures coming from nearby GPS addresses, making up to 31 clusters, indicating there are that many touristic spots featured in the data. Each of these clusters denotes a particular location of the town. The other clustering algorithm is a Mean-Shift algorithm that measures visual similarity between pictures [38]. This step allows us to identify several subclusters, which denote different viewpoints of the same place. Four annotators sorted each of these subcluster's pictures attending to how representative of the cluster and the location they were. From this arrangement, the top 10 images from the top 10 subclusters of each of the 31 clusters are taken. Not all the clusters had that many examples, so this process yields a total of 984 images that conform the final dataset. The details of the data retrieval methods introduced in this section can be found in [39].

### 3.2. Labeling Procedure

A collaborative annotation process was carried out online. Each user was presented with 50 randomly selected images, each along with the question of whether it was suitable or not, in terms of aesthetics, for the touristic promotion of the town (e.g., by including it in a touristic guide). A total of 232 people participated, so each image has on average about 12 annotations. In Figure 3, we include the histogram of scores for all the images. Scores are computed as the ratio of accepting votes to the total number of votes, i.e. number of people who considered the image as suitable divided by the total number of people that were presented that image. Then, scores are normalized to lie between 0 and 10.

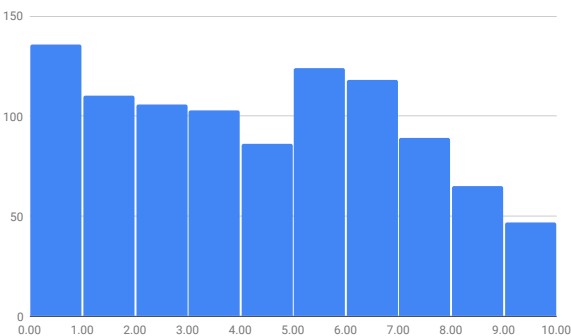

**Figure 3.** Histogram of scores in the ESITUR corpus.

The bulk of the annotations is concentrated around the central part of the histogram, like in other picture aesthetics datasets such as AVA [40]. This can be explained because most of the human reactions to the photographs are mild, not showing either great enthusiasm or displease. The agreement level on the annotation process, estimated by computing the Krippendorff's alpha coefficient [41], took a value of 0.3 showing a relatively low agreement among annotators. This proves the strong subjectivity that is implicit to the task of assessing aesthetics in tourism-oriented environments, making it particularly challenging. In fact, the amount of images that show a clear agreement is small (only 47 images scored 9 or higher). Given the purpose of our annotation (the suitability of an image for tourism applications), the labeling is not only driven by the aesthetics of the image. Instead, factors related to the fame or the content of a picture, despite the intrinsic aesthetic experience it evokes, make the final distribution look quite different from the distributions observed in other datasets [40,42].

*3.3. Segmentation of the Corpus*

Given that the data has been labeled in terms of touristic attractiveness, the score of an image is not only driven by the aesthetics of its visual features, but also by how representative of a place it is. This is an essential additional information. Because aesthetics score distribution in ESITUR is not strictly dependent on the particular aesthetics of the pictures but rather on the object or place they depict, we proceeded to compute the location average score: Each image contains a label indicating the location it belongs to, so we can estimate the location aesthetic score as the mean value of the images' of that location (Figure 4). It turns out that we can identify 3 groups of data driven by the location average score: First, locations that are regarded as unsuitable for touristic promotion (those with a score lower than 4). Second, those locations that look nice to most users and have an average score above 5.5. In between, there is a set of locations that evoke mild reactions and have location average scores between 4 and 5.5. However, it is fundamental to point out that in all these sets of locations, the score of a particular picture can have any value. From a practical point of view, we nonetheless seek to clearly tell unsuitable images, (those with a score lower than 4) from those that look nice to most users (that have an average score greater than 5.5). This estimation is highly dependent on the location a image depicts. Thus we treat the problem as a classification problem, in which the objective is to tell apart the images based on their individual score, but supported by a data partition in three data segments. This motivates a split of the corpus in three location segments given by the average score of the different locations studied: Bottom (score smaller than 4.0), middle (score between 4.0 and 5.5) and top (score greater than 5.5).

| BOTTOM | 13 Locations | | | | | | | | | | | | |
|---|---|---|---|---|---|---|---|---|---|---|---|---|---|
| Location index | 1 | 2 | 3 | 4 | 5 | 6 | 7 | 8 | 9 | 10 | 11 | 12 | 13 |
| Location average score | 0.7 | 1.3 | 1.6 | 1.9 | 2 | 2.4 | 2.8 | 3 | 3.1 | 3.2 | 3.3 | 3.7 | 3.7 |
| # Images per location | 19 | 42 | 28 | 93 | 11 | 23 | 44 | 32 | 70 | 16 | 15 | 22 | 9 |

| MIDDLE | 10 locations | | | | | | | | | |
|---|---|---|---|---|---|---|---|---|---|---|
| Location index | 14 | 15 | 16 | 17 | 18 | 19 | 20 | 21 | 22 | 23 |
| Location average score | 4.82 | 4.85 | 4.9 | 4.91 | 5 | 5.03 | 5.14 | 5.27 | 5.41 | 5.43 |
| # Images per location | 61 | 10 | 9 | 34 | 2 | 15 | 37 | 52 | 62 | 14 |

| TOP | 8 locations | | | | | | | |
|---|---|---|---|---|---|---|---|---|
| Location index | 24 | 25 | 26 | 27 | 28 | 29 | 30 | 31 |
| Location average score | 5.89 | 6.02 | 6.13 | 6.22 | 6.22 | 6.28 | 6.61 | 7.19 |
| # Images per location | 10 | 48 | 73 | 6 | 4 | 79 | 24 | 20 |

**Figure 4.** Average location score of the 31 locations found in the dataset (given by their indices within the corpus) and data split into segments accordingly: Bottom, middle and top.

An image is considered to have a high aesthetics score if at least half of the annotators considered it suitable for the touristic promotion of the town, otherwise it is tagged with a low score. Table 1 shows the amount of instances and the mean score of each data segment considered, as well as the mean difference between the images rated with high score and those rated with a low score. In all the data segments there are both photographs with high and low score, although it can also be seen that the mean score of the samples increases progressively as we go from the bottom to the top segment.

**Table 1.** ESITUR corpus segments' distribution of samples, split for both train and test sets. Indices are indicatives of a location within the data set.

| Phase | Data Segment | # Locations | # Pics. | Mean Score | Mean Diff (Low,High) | % High Aesthetics Pics. |
|---|---|---|---|---|---|---|
| Train | All | 31 | 492 | 4.22 | 4.74 | 38.41 |
| | Bottom | 13 | 203 | 2.21 | 5.02 | 9.36 |
| | Middle | 10 | 147 | 4.96 | 3.82 | 46.26 |
| | Top | 8 | 142 | 6.33 | 3.44 | 71.83 |
| Test | All | 31 | 492 | 4.23 | 4.75 | 38.62 |
| | Bottom | 13 | 221 | 2.61 | 5.11 | 16.29 |
| | Middle | 10 | 149 | 5.17 | 3.94 | 48.99 |
| | Top | 8 | 122 | 6.02 | 3.59 | 66.39 |

The difference between the mean score of high-valued images and the mean score of low-valued images reaches its peak in the bottom domain. The lower this difference, the more similar pictures rated as suitable to a touristic guide will be to those that do not. As a consequence, the task of separating them becomes more complex. This suggests that, although in general the locations in the bottom data subset are usually not regarded as aesthetically pleasant, some of them have obtained a high score and cannot be ignored by the model. This split of the data in three independent segments will define the creation of expert learners explained in Section 4.2.2.

## 4. Aesthetics Label Prediction Models and Experimental Setup

### 4.1. Feature-Based Model

First, we use a traditional approach that relies on the extraction of visual descriptors from images. These descriptors are handcrafted, meaning that they are manually designed beforehand, rather than having a model that learns which features to extract, as a neural network would do. In the rest of this section, we first describe in detail every visual descriptor, and then we present the techniques and algorithms used to create our model.

### 4.1.1. Visual Descriptors

We compute a total of 18 visual features. The capacity of these descriptors to effectively reflect image characteristics that influence its perception by humans is crucial. Consequently, we use descriptors related to properties such as color, composition, texture, etc. While most of them were used and validated previously in works on automatic perception assessment [8,43], some others, like the ones related to the horizon, are designed from scratch by ourselves. Vanishing point detection and horizon line location within an image determine the perspective of it, which in turn have an impact on the aesthetics. The visual descriptors that are used in our feature-based model are briefly explained in Table 2.

**Table 2.** Summary description of the handcrafted features computed for aesthetics estimation.

| Feature | Description |
|---------|-------------|
| Intensity | Mean brightness |
| Hue | Mean value of hue channel after transforming to HSV color space |
| Saturation | Mean value of saturation channel after transforming to HSV color space |
| Entropy | Entropy of image's pixels |
| Colorfulness | Difference between image's color histogram and an uniform color histogram |
| Color profiles | Difference between image's color histogram and a reference histogram for 8 colors |
| Rule of thirds | Measure of how consistent the horizontal lines in the image are with this composition technique |
| Horizon line | Presence and properties of an horizon line in the image, estimated using the vanishing point position. |

### 4.1.2. Feature Selection and Classification Model

Even though our descriptors have been specifically designed taking into account the purpose of modeling aesthetics, there may be some of them that do not provide any useful information for this task. Eighteen visual features are presumably not enough to describe all the aspects of an image that influence its aesthetic perception but still, it is convenient to carry out a feature selection procedure to discard those that might obstruct the learning process.

We select the K best features according to the scores obtained computing chi-squared statistics between the target class and each one of them. The number K of selected features is tuned in order to find the one that provides best results when using a simple classification algorithm. We have decided to use a basic logistic regression algorithm for classification, since this feature-based method will only serve as baseline and reference for comparison with our model based on deep learning techniques. We are aware that the exploration of classifiers that incorporate additional information such as location could lead to better results using this same set of visual descriptors.

### 4.2. Deep Learning Approach

Contrarily to hand-crafted feature extraction, neural networks are able to automatically extract a dense representation of the visual stimuli. We use an two network architectures: Alexnet, made up by 5 convolutions followed by 3 fully connected layers [44], and a VGG-19 with batch normalization architecture [45]. The latter is composed of 19 convolutional layers of different kernel size with ReLU activations [46] and batch normalization [47] after each layer, followed by a final fully connected layer.

After a fine-tuning of the hyperparameters, we found optimal results for a learning rate decay of $\gamma = 0.98$, learning rate 0.001, with batch size 64. The rest of hyperparameters are the default ones when downloaded from Pytorch framework's repository, which is the framework we use [48]. We update the weights of the models following a Stochastic Gradient Descent [49] with backpropagation as the training algorithm, using an early stopping strategy so the training is interrupted after 15 epochs without improvement in the accuracy over the validation set, to prevent overfitting. No data augmentation procedure is applied since any change in the images would affect its aesthetics. However, all pictures are converted to $224 \times 224$ pixels before feeding them to the neural network.

### 4.2.1. Canonical Model

Our deep learning baseline approach consists of models pretrained on Imagenet [44] or in AVA [40], and thus knowing nothing about data distribution in ESITUR. Transfer learning is a training procedure that allows for an efficient retraining of neural networks when the amount of data would be insufficient to effectively train the net from scratch on the task of interest [50]. This is indeed our case, since the total number of samples in our corpus accounts up to just 984 images. A model pretrained on Imagenet or in AVA is thereafter fine-tuned over the ESITUR data. The result is a canonical model, and it conforms the initialization point for the mixture of experts we explain next.

### 4.2.2. Model-Wise Mixture of Experts

The domain-wise adaptation strategy has proved to be a successful approach in aesthetics prediction [17]. However, instead of strictly following some intuition-based rules of photography to conform our experts creation, we have defined 3 data segments based on the location of the images, as described in Section 3.3. Hence, for the ESITUR corpus three different splits are considered.

The training scheme of this combination of model-wise experts is depicted in Figure 5; each expert is fine-tuned from the canonical model on the specific segment of data it is associated with, thus providing us with three different models. All the models are trained separately to avoid unwanted learning correlations [51,52] since we want them to be independent experts.

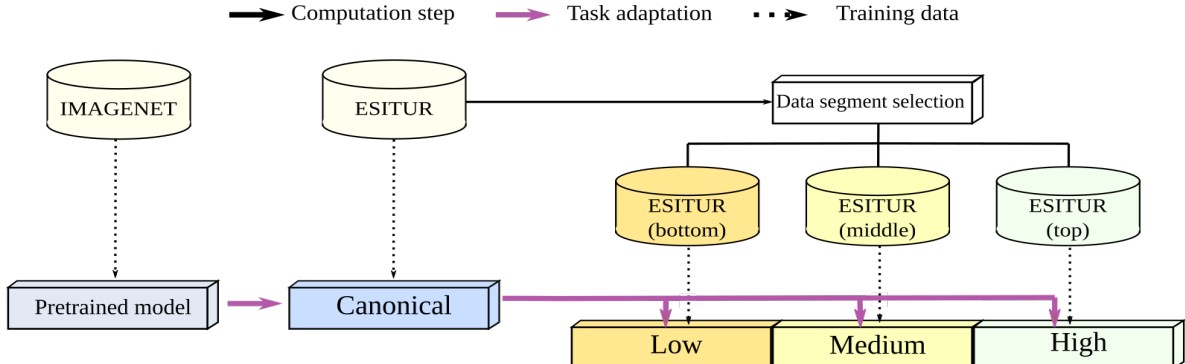

**Figure 5.** Traning phase scheme for domain adaptation experts. Ultimately we end up having 3 different, adapted models, depending on the origin of the input image.

In test time, the pipeline is shown in Figure 6. The data segment selection part acts as the gating network in conventional mixture of experts approaches, given that each data segment has a unique model expert associated. In the ESITUR corpus all the images are assigned a precise location (Section 3.1), so it is feasible to perfectly select an expert for each input image attending to which data segment its location belongs to. In case of missing labels, a possible way to scale up our solution could be to rely on an external model trained in scene categorization so input images would be automatically segmented.

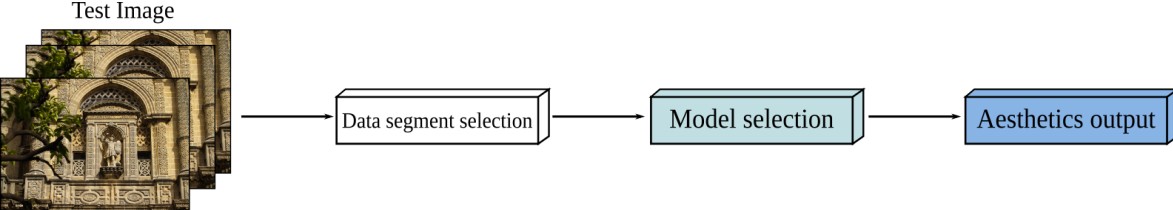

**Figure 6.** Test phase scheme for domain adaptation experts.

Models trained on datasets such as Places365 [53] offer a reasonable starting point for mixture of experts approaches similar to the one we propose here. In the experiments involving deep neural networks, the setup considered is a stratified 10-fold cross validation over the whole ESITUR corpus [54], while experiments on the handcrafted features were performed using a uniformly distributed 50%/50% train/test split of the data.

## 5. Results

### 5.1. Hand-Crafted Features

First, we present the results when employing our feature-based model both for AVA2 and ESITUR datasets. The AVA2 dataset is a reduced version of the original AVA distribution first used in Jin, X. et al. [42]. Instead of considering all data, only the top 10% of both edges of the distribution are taken, yielding samples that are either clearly aesthetically pleasant or disgusting. Therefore, it can be regarded as a simplified version of the data that allows us to easily train and evaluate our low-level, feature-based model. The feature selection process shows that all the features were valuable to solve the classification task when using AVA2. In Figure 7a, it can be seen how both train and test accuracy plots have an increasing trend with respect to the number of selected features. While choosing always one class (since half of the set is tagged as positive and the other half as negative) would yield a $50.00 \pm 0.43\%$ accuracy, the highest accuracy obtained is $60.09 \pm 0.60\%$ and $59.6 \pm 0.61\%$ for train and test splits, respectively. This suggests 18 visual features are not enough to fully model an image from the point of view of aesthetics perception.

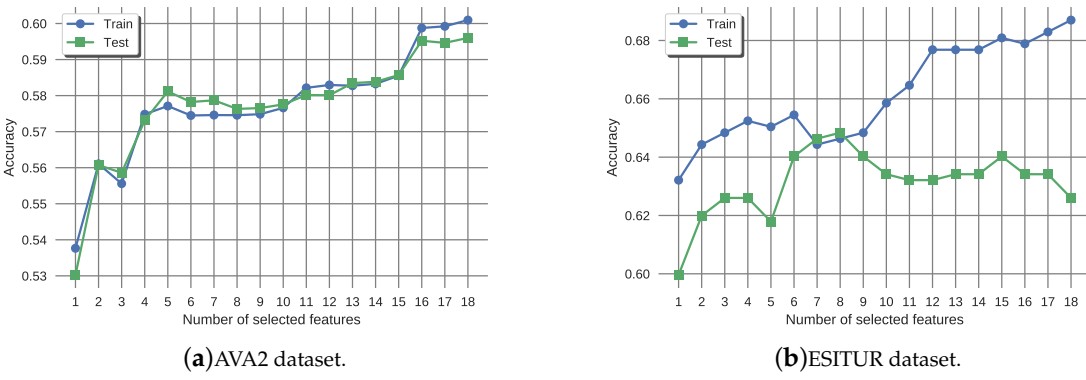

(**a**) AVA2 dataset.    (**b**) ESITUR dataset.

**Figure 7.** Accuracy rates for the train and test splits when varying the number *K* of selected features.

However in the case of the ESITUR dataset, the best test results are achieved when discarding more than half the total number of features. In Figure 7b, we plot the evolution of the accuracy value for the train and test splits when varying the number of selected features. As illustrated in the figure, the best result is obtained when using 8 features, reaching a test accuracy of $64.84 \pm 4.22\%$, only slightly above from the zeroR classification (choosing the majority class) of $61.48 \pm 3.04$. Again, this result suggests that the gain margin with this kind of approaches is limited, barely improving upon the baseline.

Since some features are discarded for ESITUR images, we consider relevant to analyze which ones led to the best accuracy result. These features were: Intensity, entropy, colourfulness, red color profile, lower and upper rule of thirds measure, and two horizon-related features. We can see that the final best selection of features is highly varied, being present low-level features such as intensity and entropy, color-related features, and, predominantly, features related to composition: The presence and position of the horizon and the use of the rule of thirds. This is not a surprising result, given that the relation between the aesthetics of an image similar high-level describable features is already well studied in the literature [8,10,31]. What is worth commenting, however, is that results on the larger AVA2 dataset lead to worse accuracy values when compared to the accuracy over ESITUR data.

This can be due to the fact that models based on describable visual features offer just a partial solution to a significantly complex problem, and AVA2 displays a much larger range of environments, topics and locations than ESITUR.

## 5.2. Deep Convolutional Models

In this section, results on the experiments carried out using neural networks are presented. Model's architecture, as well as the pretrained weights on Imagenet are downloaded from the official Pytorch's repository. By training models on AVA and using them as a pretrained initialization point for other models, we are compelled to assess their performance before actually adapting those models to any other domain. In Table 3, a list of the accuracy rates obtained with some of the most relevant systems in the literature are shown. Low-level feature-based models achieve low accuracy compared to data-driven convolutional neural networks approaches, and hence, in light of our results using low-level handcrafted features (Section 5.1), we focus on deep convolutional networks for the rest of this study. It can be noticed that our pretrained models do not reach state-of-the-art performance. It is not a concern since we are only interested in having a simple model easily adaptable to our domain of interest (ESITUR). Therefore, the pretrained canonical models we consider here suggest reasonable initialization points, given that their performance is considerable better than low-level feature-based models, yet not too far from deep learning models specifically designed for another aesthetics evaluation problem.

**Table 3.** Results available in the literature using the complete AVA dataset.

| Strategy | Model | Accuracy on AVA$_{\delta=0}$(%) $\pm$ 95% Conf. |
|---|---|---|
| Low-Level Features | Murray et al. [40] | $66.70 \pm 0.18$ |
| | Marchesotti et al. [31] | $68.55 \pm 0.18$ |
| | Datta et al. [8] | $68.67 \pm 0.18$ |
| | Ke et al. [10] | $71.06 \pm 0.18$ |
| Convolutional Networks | Alexnet (pretrained) | $74.04 \pm 0.17$ |
| | VGG-19 (pretrained) | $77.59 \pm 0.16$ |
| | Talebi, H. & Milanfar, P. [19] | $80.60 \pm 0.15$ |
| | Ma et al. [16] | $82.5 \pm 0.15$ |

Table 4 shows the accuracy rates over the ESITUR corpus obtained by the different approaches considered: Pretrained models directly applied on ESITUR, models once they are adapted to the ESITUR domain and MEXs. It is noticeable that, with no exception, the pretrained models without domain adaptation perform even worse than a simple zeroR (classifier that predicts the majority class) that yields a baseline accuracy of $61.48 \pm 3.04$ over ESITUR data. This result highlights the differences between our corpus and other aesthetically annotated datasets (as explained in Section 3.2), and the need to develop models adapted to the task at hand.

In both Alexnet and VGG architectures the experiments show similar results, this is, that in all cases the MEX approaches outperform their canonical counterparts. It is relevant to analyze the results obtained with the Alexnet network, the smallest one studied, which has about 57 M parameters (VGG has about 139 M parameters). A MEX approach enables the experts to take advantage of being allowed to specialize in just a small subset of the data, concentrating all its efforts in learning the main features within that subset. We reckon that a larger dataset would allow the architecture to better learn more complex features present in the images. These results suggest that small models may benefit the most from this MEX approach, particularly when they cannot afford to learn the complete data distribution but rather smaller, homogeneous partitions of it.

**Table 4.** Aesthetics tag prediction accuracy over the ESITUR dataset for the models considered in this study. We distinguish between aesthetic models pre-trained on AVA but with no adaptation to ESITUR, adapted canonical models and MEX models (derived from adapted canonical models).

| Model Type | Model Number | Model Training Data | Model Adaptation (Starting Model) | Accuracy $\pm$ 95% Interval (%) | |
|---|---|---|---|---|---|
| | | | | AlexNet | VGG-19 |
| Regular pre-trained models | 1 | AVA$_{\delta=0}$ | - | 45.7 $\pm$ 3.11 | 46.29 $\pm$ 3.11 |
| (no ESITUR adaptation) | 2 | AVA$_{\delta=1}$ | - | 45.7 $\pm$ 3.11 | 50.39 $\pm$ 3.12 |
| Canonical | 3 | ESITUR | Model 1 | 77.64 $\pm$ 2.6 | 76.51 $\pm$ 2.65 |
| (adapted to ESITUR from | 4 | ESITUR | Model 2 | 77.76 $\pm$ 2.6 | 76.92 $\pm$ 2.63 |
| pre-trained models) | 5 | ESITUR | Imagenet pre-trained | 82.02 $\pm$ 2.4 | 81.46 $\pm$ 2.43 |
| MEX | 6 | ESITUR | Model 3 | 80.72 $\pm$ 2.46 | 80.37 $\pm$ 2.48 |
| (adapted from | 7 | ESITUR | Model 4 | 83.53 $\pm$ 2.32 | 79.5 $\pm$ 2.52 |
| canonical models) | 8 | ESITUR | Model 5 | 84.27 $\pm$ 2.27 | 85.08 $\pm$ 2.23 |

In order to evaluate the performance on the MEX approach, we carry out an ablation study (Table 5). All the experts become specialist learners on their subset of data, outperforming the canonical model. However, due to that specialization, the learners also get worse at different data segments that the ones they are trained from. Such learners' specialization is shown in Figure 8, where the activation maps of the last convolutional layer of the network are superimposed to the original image to that particular image. Red areas denote parts of the image that make the model predict a high aesthetics value, while blue or purple ones mark those with low aesthetics. For each input image, we can see that each expert has learned to focus its response on different regions of the image depending on the picture's content. The canonical model shows a combination of the experts' learned features. Given the particular nature of our dataset, having different learners that can specialize in particularly relevant features within touristic images allows the model to increase the accuracy in telling guide-suitable pictures from the ones to reject.

**Table 5.** Ablation study of the accuracy (%) of the canonical and expert VGG-19 models over all the segments of the ESITUR corpus.

| Initialization | Data Segment | Model | | | | |
|---|---|---|---|---|---|---|
| | | Low | Medium | High | Canonical | zeroR |
| AVA$_{\delta=0}$ | All | 76.52 | 75.29 | 77.32 | **76.51** | 61.38 |
| | Bottom | **91.52** | 84.91 | 88.46 | 88.7 | 83.71 |
| | Middle | 70.03 | **74.93** | 69.23 | 71.51 | 51.01 |
| | Top | 60.81 | 61.84 | **68.58** | 63.5 | 66.39 |
| AVA$_{\delta=1}$ | All | 76.72 | 76.51 | 76.4 | **76.92** | 61.38 |
| | Bottom | **90.84** | 87.07 | 86.81 | 88.96 | 83.71 |
| | Middle | 69.62 | **74.53** | 70.36 | 73.03 | 51.01 |
| | Top | 62.78 | 63.1 | **66.85** | 63.1 | 66.39 |
| Imagenet | All | 77.71 | 76.53 | 78.91 | **81.46** | 61.38 |
| | Bottom | **92.95** | 79.85 | 84.45 | 90.11 | 83.71 |
| | Middle | 64.54 | **81.8** | 73.13 | 77.71 | 51.01 |
| | Top | 67.67 | 67.06 | **76.12** | 72.43 | 66.39 |

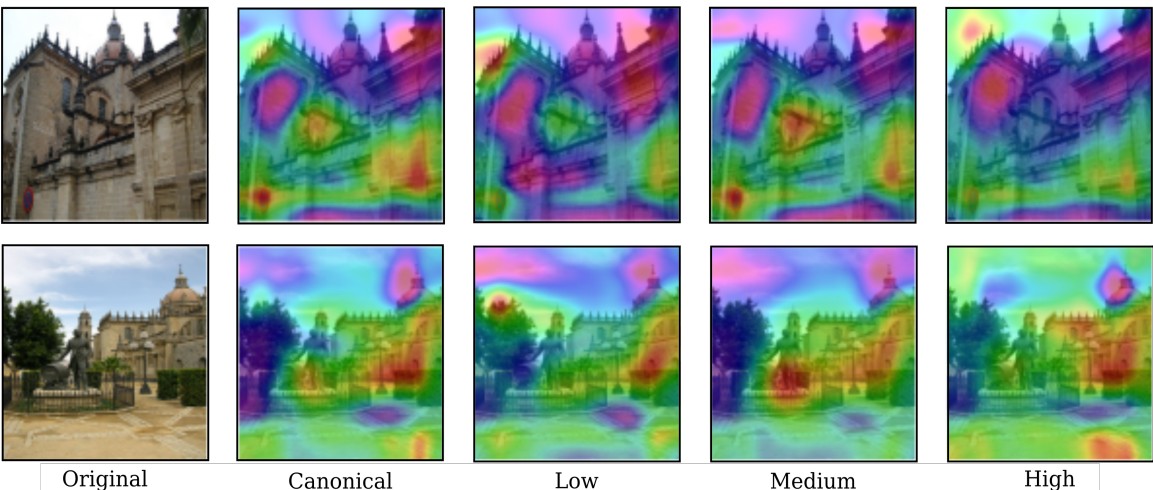

|       |       |       |        |      |
|-------|-------|-------|--------|------|
| Original | Canonical | Low | Medium | High |

**Figure 8.** Example of VGG-19-like experts' different feature activation maps after the last convolutional layer on a bottom-image (**upper**) and a top-image (**down**) in the ESITUR corpus.

## 6. Conclusions

Convolutional neural networks have proved to be a more accurate solution to the problem of giving an aesthetics label to an image than handcrafted visual features, particularly when considering the suitability of those pictures to tourism applications. Whereas handcrafted features seem to reach an upper limit hard to overcome, neural approaches generalize much better to unseen images. Even more, handcrafted approaches have been unable to scale up to larger sets of data with a wider variaty of perspectives and environments, showing a performance only slightly above the zeroR classification in both small and large sets of data.

We have sought data-driven neural experts whose architectures (Alexnet and VGG-19 with batch normalization) are well known, in order to fully explore the benefits purely derived from the Mixture of Experts approach rather than from the complexity of the network. We have shown that a noticeable improvement can be achieved independently of the model architecture, encouraging research towards dense models that can be trained quickly in a real-world application.

In particular, our results show that a single global model that spans the whole dataset is not an option as good as a Mixture of Experts whose gating function is driven by implicit features present in the data. In our study, that additional information comes from the location a given picture depicts, which is automatically computed from the GPS coordinates associated with it.

As future work we leave the exploration of turning the task into a regression one. With the current approach, an image is either accepted or rejected. This can be troublesome in the case of images with a very weak acceptance score (cases for which the number of people who liked it is roughly equal to the number of those who disliked it), since once its aesthetics value is classified, it will be indistinguishable from another with a higher score. In addition, we are interested in scaling up the Mixture of Experts approach to new gating strategies, based on attributes different from the location of the input image. Such attributes can come from other metadata commonly available in social media such as interaction with the content.

**Author Contributions:** Conceptualization, R.K., J.M.M. and F.F.-M.; methodology, R.K., J.M.M. and F.F.-M.; software, R.K., Á.G.-F., C.L.J. and F.F.-M.; validation, R.K., Á.G.-F., C.L.J. and F.F.-M.; formal analysis, R.K., Á.G.-F., C.L.J. and F.F.-M.; investigation, R.K., Á.G.-F., C.L.J. and F.F.-M.; resources, F.D.-d.-M.; data curation, R.K., Á.G.-F. and F.F.-M.; writing—original draft preparation, R.K. and Á.G.-F.; writing—review and editing, R.K. and F.F.-M.; visualization, R.K., Á.G.-F. and F.F.-M; supervision, J.M.M. and F.F.-M.; project administration, F.D.-d.-M. and F.F.-M.; funding acquisition, F.D.-d.-M. and F.F.-M.

**Funding:** The work leading to these results has been supported by the Spanish Ministry of Economy, Industry and Competitiveness through the ESITUR (MINECO, RTC-2016-5305-7), CAVIAR (MINECO, TEC2017-84593-C2-1-R), and AMIC (MINECO, TIN2017-85854-C4-4-R) projects (AEI/FEDER, UE).

**Acknowledgments:** We gratefully acknowledge the support of NVIDIA Corporation with the donation of the Titan X Pascal GPU used for part of this research.

**Conflicts of Interest:** The authors declare no conflict of interest.

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
