# Peer review of "Predicting Image Aesthetics for Intelligent Tourism Information Systems"

_electronics, doi:10.3390/electronics8060671_

Round 1
Reviewer 1 Report
This paper presents an interesting framework for estimating the aesthetics of an given image.
My comments and concerns are,
1) It seems very difficult to quantify the aesthetics even by a bunch of people, and it is not clear how biased the labeling process could be. please give more explanation, such as, are all the people labeling those images from local, or it doesnt matter where they are from or they have been to the actual places.
2) Please gives some more discussion about how the framework can be implemented so that the label could be updated by real-time scores, for example users may give different scores from time to time, which makes the whole algorithms more solid.
3) It is a little bit confusing when the author is talking about the location scores, please clarify the following: i) how many locations there are, ii) why category by score 4 and 5.5 (why not 4 and 6), iii) how many images per location, iv) is the location average the mean of scores for each location, v) what does it mean by half of the annotators considered suitable, if the task is binary classification, why using the scoring system, it could be just a binary classification, because the author mentioned later that the future work could make it a regression problem.
4) Not sure what is the data segment, does it have anything to do with the score threshold of 4 and 5.5? Please gives more clearer explanation, and also why is that important.
Author Response
@page { margin: 0.79in } p { margin-bottom: 0.1in; direction: ltr; line-height: 115%; text-align: left; orphans: 2; widows: 2 }Point 1: It seems very difficult to quantify the aesthetics even by a bunch of people, and it is not clear how biased the labeling process could be. Please give more explanation, such as, are all the people labeling those images from local, or it doesn’t matter where they are from or they have been to the actual places.
Response 1: Thank you, this is a question of fundamental importance in the definition of our problem. While it is true that the process of quantifying the aesthetics of an image is prone to be biased or influenced by personal previous experiences, the aesthetics label we consider must be understood in terms of aesthetics value for an interactive touristic guide of a town. This is a crucial aspect given that in tourism environments, pictures depicting places with a high implicit value (such as cathedrals, monuments and famous landscapes) usually happen to be considered better for touristic promotion, leaving the actual quality of the picture to a second place.Moreover, local annotators would effectively introduce a biased opinion towards particular places due to their emotions or personal experiences despite the actual suitability of the image for touristic promotion or its quality. In order to address that issue, the recruited annotators are all from the same country and, although some of them may have heard of it before, they have never actually visited the town of interest. Therefore the labeling of a sample is driven by both its aesthetic value but also by how interesting in terms of tourism the image is, as seen by people with no prior personal attachments to the town. We consider this selection the best way to reduce the possible bias related to choosing appropriate pictures for the promotion of a city.
Point 2: Please give some more discussion about how the framework can be implemented so that the label could be updated by real-time scores, for example users may give different scores from time to time, which makes the whole algorithm more solid.
Response 2:We have included an appendix describing the main features of the complete framework, where real-time scoring and user interaction is taken into account in order to dynamically build interactive touristic showcases from multimedia content available in social media. In such scenario, new images are uploaded, and old ones receive new feedback as a result of the users’ interaction. This dynamic procedure of data retrieval is fully described in a recently published paper (Pla-Sacristan et al. 2019)we now cite in our paper. A regularly-settled updating of the corpus opens the door for the creation of new expert models, or the fine-tuning of old ones, based on the incoming images and metadata uploaded to the social media by the users. The study presented in our work is oriented towards the aesthetics evaluation of the content provided from that data retrieval pipeline. Hence, we carry out an experimentation process as close as possible to the final use case ofour model, using images from a specific town that could be the subject of an integral intelligent tourism solution.
Point3:It is a little bit confusing when the author is talking about the location scores, please clarify the following: I) how many locations there are,ii) why category by score 4 and 5.5 (why not 4 and 6),iii) how many images per location, iv) is the location average the mean of scores for each location,v) what does it mean by half of the annotators considered suitable, if the task is binary classification, why using the scoring system, it could be just a binary classification, because the author mentioned later that the future work could make it a regression problem.
Response3:Because we reckon this section was not as clear as we expected, we have changed Figure 4 and Table 1, in the hope it will help understanding of the concept of location scores. Also, additional changes have been made:
i) We have rephrased the text in section 3.1 in order to make it clearer there are 31 touristic spots which are equivalent to having 31 different locations.
ii) Our predictive systems rely on defining homogeneous segments on the data. Without such homogeneity, the expert predictive models would be unbalanced and we would not take fully advantage of the experts’ ability to extract patterns over specific groups of data. We have chosen thresholds so such data segments are semantically meaningful while preserving the uniformity of the data distribution.
iii) We have included that information in both Figure 3.
iv) Yes, it is exactly the mean of scores of all the images for each location.
v) An image is considered to have a high aesthetics score (1) if at least half of the annotators considered it suitable for the touristic promotion of the town, otherwise it is tagged with a low aesthetics tag (0). That aesthetics tag is the label to predict. The location scoring system allows us to define experts that specialize on segments of the data depending on the location. These segments denote groups of locations that on average achieve lower aesthetics value, even though in all the segments there are both images with a low aesthetics tag (0) and images with a high aesthetics tag (1), as it is shown in Table 1. Because of that, the same approach could be used to tackle a regression problem.
Point4:Not sure what is the data segment, does it have anything to do with the score threshold of 4 and 5.5? Please give more clearer explanation, and also why is that important.
Response4:We understand this question is related to the answers provided to the previous question, section ii) and v). A data segment is a subset of the whole dataset, defined by the location tags available. Locations with similar average scores are grouped together into a data segment. The thresholds proposed are sought to provide homogeneous segments accross the whole dataset.

Reviewer 2 Report
1. There are grammatical errors. Careful proof reading is suggested.
2. All the acronyms used in the paper should be explained.
3. The dataset used is comparatively small for training deep neural networks, and over fitting might be a problem.
4. The authors state that they did not do data augmentation as any change in the images would affect its aesthetics. However, the experiments are conducted with a down-sampled version of the original images. Does downsampling have any effect on the perceived aesthetics of the images? Was the annotation done using the downsampled images?
5. The authors state that early stopping was used by observing the accuracy on the validation set. However, details of the validation set is missing.
6. It is already known from prior works that deep learning models can be used for the assessment of the aesthetic quality of images. The deep learning model used in this paper does not improve on the results over the existing methods. What is the main contribution of the paper?
7. In conclusion the authors state that “ We have shown a noticeable improvement can be achieved without need for more complex architectures …”. No experimental results on comparison with other methods such as architecture complexity or training/inference time efficiency to support this claim is provided.
Author Response
@page { margin: 0.79in } p { margin-bottom: 0.1in; direction: ltr; line-height: 115%; text-align: left; orphans: 2; widows: 2 }Point 1: There are grammatical errors. Careful proof reading is suggested.
Response 1: Thank you very much. We gladly accept your indication, and we have rephrased misleading or confusing sentences.
Point 2: All the acronyms used in the paper should be explained.
Response 2: We appreciate the reviewer’s suggestion, and after a careful reviewing of the acronyms included in the paper, we have found that they refer to methods widely studied in the literature and usually referenced by their acronyms by most other authors. Hence, we understand the reader is familiar with such methods and it is not necessary to further explain their meaning.
Point 3: The dataset used is comparatively small for training deep neural networks, and over fitting might be a problem.
Response3: While it is true that the dataset we work with is limited in resources, to our humble knowledge this is the very first attempt on collecting an aesthetically annotated dataset for tourism. We agree with the reviewer that having a small dataset may make the model incur in undesirable consequences such as overfitting. Therefore, we have performed additional 10-Fold Cross validation analysis, which is usually found to be one of most important strategies to prevent overfitting, finding more solid statistical evidences (with smaller error intervals) than those presented before in the paper. Hence, we have adopted it as the standard setup for all the experiments carried out in this study.
Point 4: The authors state that they did not do data augmentation as any change in the images would affect its aesthetics. However, the experiments are conducted with a down-sampled version of the original images. Does downsampling have any effect on the perceived aesthetics of the images? Was the annotation done using the downsampled images?
Response 4: We agree on that any transform in an image affects its properties. Nonetheless, downsampling of the images is a reasonable and much needed procedure, commonly used in the deep learning community, and also in other aesthetics evaluation studies to alleviate the problem complexity, which in turn ensures model trainability (simplified models that require less computational resources) and problem tractability (Kao et al. 2016 , orTalebi, H. and Milanfar, P., 2018). Although it is true that some pixel information is lost, that loss is relatively small (the main features of the image remain untouched). Hence, downsampling ends up being a suitable trade-off between performance and complexity.
On the second question, annotators were presented with images of their actual size. The images are only resized right before feeding them to the neural network. Since downsampling does not alter the main features of the image, we do not consider the fact that human labeling was done over the original images instead of the downsampled ones as a critical issue. This happens to be the usual case in the literature.
Point 5: The authors state that early stopping was used by observing the accuracy on the validation set. However, details of the validation set is missing.
Response 5: Considering the reviewer’s comments, we have now adopted a 10-Fold Cross Validation analysis as we consider it is a good alternative to our previous training-validation scheme, making it our standard setup in the experiments presented in this study.
Point 6: It is already known from prior works that deep learning models can be used for the assessment of the aesthetic quality of images. The deep learning model used in this paper does not improve on the results over the existing methods.What is the main contribution of the paper?
Response 6: While it is true that the assessment of the aesthetic quality of images is a problem addressed in other studies, and effective methods have be en proposed for that particular problem, what we observe is that if we directly evaluate a model previously trained on another well-known aesthetics datasets such as AVA without domain adaptation to our problem, the performance of the model is poor. We have now included these results in Table 5, because it points out how different the domains of other aesthetics datasets and our corpus are. As a consequence, we see no profit in adapting a state-of-the-art method that is designed and trained with another aesthetics task in mind, since it seems unlikely that our solution will be benefited from such problem-specific methods.
Previous studies contemplate the problem of assigning an aesthetics value to input images implementing predictive architectures that fully exploit the model’s capacity to adapt to a very specific data distribution. Instead, we demonstrate that models showing a reasonable performance on a well-known corpus such as AVA can be adapted to our problem domain with success, and that an approach based on a Mixture of Experts scheme can further improve our results.
Point 7: In conclusion the authors state that “We have shown a noticeable improvement can be achieved without need for more complex architectures...”. No experimental results on comparison with other methods such as architecture complexity or training/inference time efficiency to support this claim is provided.
Response 7: We have rephrased that sentence, since we understand it was not totally clear before. Our aim was to point out that the proposed approach of dividing the data space in order to build expert models is independent of the particular model architecture used. This is supported by previous experiments we performed using a simpler Alexnet architecture, in which we observed similar results to those presented in the paper. We have included some results with both architectures, although we do not report on all the results since experimentally we found both architectures present similar results, but with Alexnet achieving lower accuracy scores. This fact has been mentioned in the text.

Round 2
Reviewer 2 Report
The authors have answered all the issues raised earlier. No further comments.